# Risk of Environmental Contamination by Gastrointestinal Parasites in Public Areas of the Central Plateau Microregion of Brazil: A Public Health Concern

**DOI:** 10.3390/pathogens14030211

**Published:** 2025-02-21

**Authors:** Ana Julia de Almeida Martins, Alice Caroline da Silva Rocha, Zara Mariana de Assis-Silva, Guilherme Oliveira Maia, Bruna Samara Alves-Ribeiro, Raiany Borges Duarte, Iago de Sá Moraes, Nicoly Ferreira de Urzedo, Lizandra Fernandes-Silva, Ana Paula Carvalho Gomes, Samara Moreira Felizarda, Mayra Parreira Oliveira, Klaus Casaro Saturnino, Hanstter Hallison Alves Rezende, Rosângela Maria Rodrigues, Dirceu Guilherme de Souza Ramos, Ísis Assis Braga

**Affiliations:** 1Laboratory of Parasitology and Veterinary Clinical Analysis, Institute of Agricultural Sciences, Federal University of Jataí, Jataí 75801-615, Brazil; martins.ana@discente.ufj.edu.br (A.J.d.A.M.); alicerocha@discente.ufj.edu.br (A.C.d.S.R.); zaramariana@discente.ufj.edu.br (Z.M.d.A.-S.); maia_guilherme@discente.ufj.edu.br (G.O.M.); raiany.duarte@ufj.edu.br (R.B.D.); iago.moraes@ufj.edu.br (I.d.S.M.); nicolyurzedo@discente.ufj.edu.br (N.F.d.U.); lizandra_fernandes@discente.ufj.edu.br (L.F.-S.); anapc_gomes@discente.ufj.edu.br (A.P.C.G.); 2Laboratory of Veterinary Anatomical Pathology, Institute of Agricultural Sciences, Federal University of Jataí, Jataí 75801-615, Brazil; brunasamara@discente.ufj.edu.br (B.S.A.-R.); klaus.sat@ufj.edu.br (K.C.S.); 3Biosciences Basic Unit, University Center of Mineiros, Mineiros 75833-130, Brazil; samaramoreira55@hotmail.com (S.M.F.); mayparreira18@gmail.com (M.P.O.); 4Laboratory of Bacteriology and Mycology, Institute of Health Sciences, Federal University of Jataí, Jataí 75801-615, Brazil; hanstterhallison@ufj.edu.br; 5Laboratory of Parasitology, Institute of Health Sciences, Federal University of Jataí, Jataí 75801-615, Brazil; rosangela_rodrigues@ufj.edu.br

**Keywords:** enteroparasites, epidemiological factors, geohelminths, One Health, Zoonosis

## Abstract

The risk of zoonotic parasitic infections is closely linked to direct and indirect interactions between animals and humans. The mutual coexistence of species in public spaces predisposes individuals to gastrointestinal parasitosis owing to various social and hygienic-sanitary factors. This study aimed to assess the risk of environmental contamination by gastrointestinal parasites in fecal samples collected from parks and public squares in 18 municipalities located in a microregion of the central plateau of Brazil, correlating the collection with the population size of each municipality. We collected 536 soil fecal samples from 117 randomly selected public areas across a 56,111.874 km^2^ region. Eggs, cysts, and oocysts were detected using the Willis flotation technique and the Hoffman’s spontaneous sedimentation method. The Hoffman’s analysis revealed that 70.3% of the fecal samples were infected across 91.5% of the sampled areas. Identified parasites included the members of the family Ancylostomatidae (56.5%), *Toxocara* spp. (6.2%), *Trichuris* spp. (1.7%), *Strongyloides* spp. (0.2%), *Dipylidium caninum* (25.8%), *Spirometra* spp. (0.4%), Taeniidae (0.2%), *Platynosomum fastosum* (0.6%), Trematoda (0.2%), *Giardia* spp. (3.2%), *Cystoisospora* spp. (5.6%), *Sarcocystis* spp. (0.2%), and *Entamoeba* spp. (2.4%). The presence of positive fecal samples in public areas and municipalities correlated with populations of up to 10,000 inhabitants (*p* = 0.023). Areas contaminated with feces were 63.4% more likely to contain parasites than other areas (odds ratio 1.6336). Favorable environmental factors combined with inadequate sanitary management contribute to a high risk of environmental contamination, representing a significant zoonotic potential and highlighting the need for improved public health policies and preventive measures.

## 1. Introduction

Public spaces, including parks and squares, provide environments for leisure and socialization, contributing to the well-being of both human life and companion animals. The movement of dogs and cats in these areas is common, whether accompanied by their owners or living as strays, which often leads to social problems, as many of these animals can serve as pathogen reservoirs [1,2,3]. Studies in Brazil and around the world demonstrate a prevalence of gastrointestinal parasites in dogs ranging from 16.1% to 62.2% and in cats ranging from 3.3% to 90.9%, according to Souza et al. [3].

Fecal contamination is one of the major concerns in these environments because feces can serve as a source of infection for both animal and human populations [4]. Previous research has demonstrated a wide variety of gastrointestinal parasites, many with zoonotic potential, in fecal material dispersed throughout the environment [5,6,7,8,9,10], but studies on environmental contamination are still scarce, especially in Brazil. In some tropical and subtropical regions, contamination is endemic due to climate and socioeconomic factors, including poor sanitation and lack of local hygiene, resulting in the persistence of parasites in the soil [11].

The high prevalence rate of fecal contamination by gastrointestinal parasites is associated with a set of factors such as the presence of many domestic animals in the country, failures in the diagnosis and treatment of animals, and insufficient sanitary hygiene measures. Brazil ranks third in the number of pets globally, with 62.2 million dogs and 30.8 million cats. Approximately 25% of these animals are abandoned, exceeding the 30 million mark of stray animals whose health is neglected, making them carriers of numerous pathogens [12].

The infective forms of zoonotic gastrointestinal parasites in dogs and cats contribute to the development of neglected tropical diseases, such as soil-transmitted helminth infections, which are among the most common infections globally, affecting 24% of the world’s population [13,14]. This scenario is particularly concerning in children and immunocompromised individuals [15].

Therefore, epidemiological studies are important to assess the risk of environmental contamination by gastrointestinal parasites in fecal samples collected from public areas to correlate possible epidemiological factors.

## 2. Materials and Methods

### 2.1. Study Area and Sampling

The study site was the southwestern microregion of Goiás, located on the central plateau of Brazil, with a territorial extension of 56,111.874 km^2^ and an estimated population of 536,973. The region includes the municipalities of Aparecida do Rio Doce, Aporé, Caiapônia, Castelândia, Chapadão do Céu, Doverlândia, Jataí (and its two districts, Estância and Naveslândia), Maurilândia, Mineiros, Montividiu, Palestina de Goiás, Perolândia, Portelândia, Rio Verde, Santa Helena, Santa Rita do Araguaia, Santo Antônio da Barra, and Serranópolis (Figure 1) [16,17]. The climate is tropical semi-humid, with an average annual temperature and precipitation of 20–28 °C and 2300 mm, respectively [16,17].

Between March 2023 and July 2024, 117 public squares and recreational parks were selected to ensure that the sampling included the central and peripheral areas of the 18 municipalities in the microregion. Visits to the cities were carried out once during the study period, according to the availability of vehicles to carry out the collections. In each collection, we went with several collection groups that divided up and went to different squares. During morning inspections lasting 15–20 min per area, fresh fecal samples were collected from the environment, avoiding portions in direct contact with the soil and dried, trampled, or rain-diluted feces. The samples were stored in collection containers, identified, and refrigerated at an average temperature of 8 °C for up to 24 h. The samples were selected when they were similar to dog and cat feces. We also observed the places where dogs and cats usually defecate in the squares.

### 2.2. Sample Processing

Fecal samples were subjected to coproparasitological techniques, including the Willis Mollay flotation method [18] and Hoffman’s spontaneous sedimentation method [19,20]. The structures were observed via optical microscopy using an Eclipse E200 instrument (Nikon, Tokyo, Japan). Eggs, cysts, and oocysts were identified using the previously established criteria [21].

### 2.3. Data Analysis

The prevalence of gastrointestinal parasites in environmental fecal samples from the sampled area was calculated using a 95% confidence interval (CI) [22]. Population size was evaluated as a risk factor for fecal contamination in public squares using the chi-square test, with statistical significance set at *p* < 0.05. Subsequently, the odds ratio (OR) was calculated for municipalities where the population size was correlated with contamination in public areas. The analyses were conducted using Epi Info™ 7.2 version software (CDC, Atlanta, GA, USA).

## 3. Results

### 3.1. Analysis of Gastrointestinal Parasites in Environmental Fecal Samples

Of the sampled areas, fecal contamination was detected in 91.5% (107/117), with 377 of the 536 fecal samples (70.3%) containing at least one immature form of a parasite (eggs, cysts, or oocysts). Table 1 presents the prevalence of fecal contamination in each municipality.

### 3.2. Diversity of Parasites Detected in Fecal Samples

Nematode eggs of the members of the family *Ancylostomatidae* (56.5%), *Toxocara* spp. (6.2%), *Trichuris* spp. (1.7%), and *Strongyloides* spp. (0.2%) were identified. Cestode eggs included *Dipylidium caninum* (25.8%), *Spirometra* spp. (0.4%), and *Taeniidae* (0.2%). Trematode eggs included *Platynosomum fastosum* (0.6%) and an unidentified Trematoda egg (0.2%). Protozoa included cysts of *Giardia* spp. (3.2%) and *Entamoeba* spp. (2.4%), and oocysts of *Cystoisospora* spp. (5.6%) and *Sarcocystis* spp. (0.2%). Table 2 presents the distribution of parasites in the sampled municipalities.

### 3.3. Risk Associated with Area Contamination

The evaluation of population size revealed a statistically significant correlation between the presence of positive fecal samples in public areas, municipalities, and districts with ≤10,000 inhabitants (*p* = 0.023) and municipalities with populations > 100,000 inhabitants (*p* < 0.001). Municipalities with populations of 10,001–30,000 and 30,001–100,000 inhabitants did not show statistically significant correlations with area contamination (*p* = 0.181 and *p* = 0.819, respectively). The OR for municipalities and districts with ≤10,000 inhabitants was 1.6336 (95% CI = 1.0671–2.501), indicating a 63.4% higher likelihood of feces containing parasitic forms in public areas in these regions than in other regions. The OR for municipalities with populations > 100,000 inhabitants was 0.5177 (95% CI = 0.3536–0.7579), indicating a 48.2% lower likelihood of feces containing parasitic forms in public areas in these regions than in other regions.

## 4. Discussion

### 4.1. High Prevalence of Parasites in Environmental Fecal Samples: An Overview of the Study Area

Our study revealed a considerable risk of environmental contamination by gastrointestinal parasites in public areas of the central plateau of Brazil, as parasites were detected in environmental fecal samples in >90% of the sampled areas. Notably, the study area in our research was 56,111.874 km^2^, exceeding the area of some countries such as Switzerland (41,290 km^2^) and Belgium (30,519 km^2^) and more than half the area of Portugal (92,090 km^2^). Therefore, the present study area is sufficient to encompass vast ecological and socioeconomic diversity. Hence, our results provide valuable data for the development of more effective public policies to control and prevent zoonoses in areas with territorial, environmental, socioeconomic, and population diversity.

Clinical epidemiological data have shown that subclinical infections from gastrointestinal parasites are widespread in domestic populations of dogs and cats [3,23,24,25]. The absence of clinical signs hinders the early identification of parasitism and proper and supervised treatment. In particular, the indiscriminate administration of antiparasitic medications to pets without veterinary guidance is common among some owners, leading to failure in parasite elimination and the development of resistance mechanisms, making these animals persistent sources of infection [26,27,28,29,30].

Inadequate management of environmental sanitation is associated with parasite contamination in urban areas [31]. The use of disinfectants based on sodium hypochlorite, quaternary ammonia, and phenols for environmental disinfection is effective [32]. Veterinarians play an important role at this stage, prescribing products and raising awareness among pet owners about the importance of interrupting the life cycle of parasites in indoor environments [3].

In a previous study, the use of a collar and the owner’s guidance reduced the presence of dog fecal samples in urban parks in the city of Calgary [4]. This strategy could serve as a preventive measure to reduce environmental contamination, alongside proper feces removal and disposal. Interestingly, in the present study, only one area (1/117) had a policy intended to raise awareness regarding the disposal of animal feces. Furthermore, few of the studied areas had physical containment in recreational areas, such as sand courts and playgrounds, to reduce access to stray animals, minimizing the risk of environmental contamination (Figure 2). There are no government notices for feces to be collected and thrown in the trash; consequently, no penalties are applied to pet owners.

Dogs and cats are considered to be the causative agents of environmental contamination due to the dissemination of geohelminth eggs with zoonotic potential [33]. Studies in Brazil [2,3] already point to this reality through the diagnosis of zoonotic parasites in feces collected from animals, a concern also evident in a similar study in Poland [34] with the detection of zoonotic species in these animals, similar to our study, such as Ancylostomatidae and *Toxocara*. Environmental studies are still scarce, but they equally demonstrate the importance of this environmental approach. Most studies on environmental contamination indicate the occurrence of Ancylostomatidae and *Toxocara* in Africa, the Americas, Asia, and Europe [11], with very few approaches that focus on a broader diagnosis and on important but neglected pathogens, such as Echinococcus and Taenia, described in environmental contamination in Iran [35]. This is partly related to the use of only one copro-parasitological diagnostic technique or the low number of samples per study. Our study focused on a robust sampling design to identify the greatest possible diversity of pathogens, in addition to the use of sedimentation and flotation techniques that allowed the identification of high and low-density eggs and oocysts.

Another notable factor is the climatic characteristics of the region, as the tropical climate provides a favorable environment for the viability of these parasites, prolonging their infectivity and increasing transmission risk. Soil moisture and the frequency of precipitation are crucial for preventing the desiccation of eggs and oocysts, whereas moderately high temperatures accelerate embryonic development, favoring infectivity [36]. Although our collections occurred at different times of the year, due to the territorial extension covered, the Brazilian Cerrado is a biome characterized by little impact by seasonal changes as in temperate zone biomes. It is noted that the temperature variation is small [16,17] compared to areas where we clearly perceive summer, autumn, winter, and spring.

### 4.2. Diversity of Parasites in Environmental Fecal Samples and Health Impacts

The range and variety of the identified parasites with zoonotic implications were notable, with the exception of *Cystoisospora* spp. and *Platynosomum fastosum*, which are associated only with infections in companion animals [3,37,38,39].

The members of Ancylostomatidae were the most prevalent geohelminth, identified in 56.5% of fecal samples. The risk of environmental contamination by this parasite extends beyond the risk of cutaneous larva migrans development from *Ancylostoma braziliense* and *Ancylostoma caninum* in humans, as the latter has been identified in fecal samples in its adult form, causing eosinophilic enteritis [40,41]. *Ancylostoma ceylanicum* has been reported in South American countries that were previously considered free from this parasite. It is an emerging public health threat capable of causing anemia due to iron deficiency and malnutrition, particularly in children [42].

Despite lower prevalence rates, fecal contamination by *Toxocara* spp., *Trichuris* spp., and *Strongyloides* spp. should not be neglected, as they can lead to significant geohelminthiasis, specifically in children and immunocompromised individuals [14]. Antonopoulos et al. described no evidence of a correlation between estimated soil contamination and the seroprevalence of toxocariasis in humans. However, a positive correlation was observed between *Toxocara* spp. prevalence in cats and dogs and seroprevalence in humans [43]. This finding indicates a public health concern, as the global prevalence in dogs is estimated at 11.1% [44], and the disease is often underdiagnosed in the human population owing to the nonspecific nature of its clinical presentation, which can vary from ophthalmological to systemic disorders. *Toxocara* spp. has also been associated with cognitive impairment in school-age children [43].

The risk of environmental contamination by *Trichuris* species is likely associated with *Trichuris vulpis*. Dogs are the definitive host of *T. vulpis*, which has zoonotic potential. *T. vulpis* has also been associated with visceral larva migrans [45]. The definite hosts of *T. trichiura* are humans and primates, and it has rarely been reported in dog feces. Its zoonotic transmission risk requires further investigation, as it is considered the second most prevalent parasite in human geohelmintiasis worldwide [46]. Strongyloidiasis, caused by *Strongyloides stercoralis*, is another neglected tropical disease common in tropical and subtropical zones. Although the prevalence of *S. stercoralis* in this study was 0.2%, its relevance should be considered with respect to global prevalence data demonstrating 6%, 13.3%, and 8.1% infection in dogs, cats, and humans, respectively [47,48,49].

*Dipylidium caninum* was identified in 25.8% of the contaminated fecal samples. The presence of *D. caninum* egg capsules in the soil and cohabitation with companion animals represents an indirect risk of infection to humans, as the infective form is only present in the intermediate hosts, such as fleas and lice. These hosts can be accidentally ingested due to poor hygiene habits and activities on contaminated ground, such as playing and eating. However, infection is uncommon in humans and primarily occurs in children in an asymptomatic form with gastrointestinal clinical presentations [50].

Considering the morphological similarity and limitations of the technique used, the eggs identified from the Taeniidae family may belong to the genus *Taenia*, whose species identified in dogs and cats do not pose a zoonotic risk, and *Echinococcus* spp., which include zoonotic species causing cystic and alveolar echinococcosis. The high molecular prevalence of canine echinococcosis in South America (66.03%) reinforces concerns about the risk of environmental contamination [51,52].

The prevalence of fecal contamination by *Giardia* spp. cysts in the environmental samples in this study (3.2%) corresponds to <10% global prevalence of infections by *Giardia* in dogs and cats, which may, however, vary according to the socioeconomic characteristics of the study area [53]. The zoonotic potential of *Giardia duodenalis* has been extensively reviewed, as molecular investigations and characterization of a variety of distinct genotypes have revealed patterns of circulation of these genotypes in animals and humans. In Brazil, giardiasis in humans is neglected, reaching extremely high prevalence rates, above 78% in the 1990s, occurring in adults and children, and with a high dispersion of cysts in the water of several cities [54]. The current hypotheses suggest that the genotypes of *G. duodenalis* predominant in dogs (C and D) and cats (F) pose a low zoonotic risk compared to those identified in non-human primates, horses, rabbits, guinea pigs, chinchillas, and beavers. Genotypes A and B that have been related to human infections were identified infecting dogs and cats, respectively. Considering that transmission of this pathogen is associated with food and water contaminated with cysts [54], and that zoonotic genotypes have been described in dogs and cats, environmental contamination with *Giardia* spp. cysts is a problem for two reasons: (1) in recreational areas such as squares and parks, people sit on the ground and on lawns, as well as children playing in these areas or having picnics, which can contaminate hands and generate infections; (2) *Giardia* spp. cysts can be carried by rain, reach water collections (rivers, streams, and lakes) that are used for human activities, and also generate infections. Although some authors have associated these findings with coprophagic habits and spillback phenomena, the zoonotic risk should not be ignored [53,55].

Cysts of *Entamoeba* spp. in environmental fecal samples may pose risks to animal and human health. Specifically, *Entamoeba histolytica* cysts are not only zoonotic but also pathogenic, causing severe clinical conditions in humans, such as bloody diarrhea, abdominal cramps, and liver abscesses [56]. While environmental contamination favors the transmission of this protozoan to susceptible hosts via its direct lifecycle, other parasites identified in this study, such as *Sarcocystis* spp. and *Spirometra* spp., have a low direct zoonotic risk, as their transmission is associated with the ingestion of intermediate hosts [57,58]. However, the presence of these two pathogens indicates circulation in the study area and a potential zoonotic risk.

### 4.3. Perspectives on Public Policies in One Health Education

The risk of environmental contamination by gastrointestinal parasites is a significant public health threat, especially in recreational areas such as public squares and parks [5,6,7,8,9,10]. The neglect of responsible pet ownership, allowing unrestricted access to streets, failures in sanitary management, uncontrolled birth rates, and animal abandonment are detrimental to animal and human health, as these animals can serve as reservoirs for various pathogens [59]. Laws in Brazil are intended to ensure the integrity and well-being of companion animals, with detention penalties and fines for non-compliance [60,61,62]. However, many pet owners do not practice responsible pet ownership [3].

Teixeira et al. highlighted the direct relationship between the lack of basic sanitation and the spread of gastrointestinal parasites in Brazil. Associated risk factors included contaminated soil, vulnerability of minority groups, housing conditions, lack of water, poor water quality used for drinking, and absence of sewage systems [63].

The risk analysis performed in the present study revealed a higher risk of environmental contamination by gastrointestinal parasites in areas with ≤10,000 inhabitants and a lower risk in areas with >100,000 inhabitants. These findings indicate the necessity to address socioeconomic differences involving sanitation and health education issues. In Brazil, most small municipalities have inferior basic sanitation infrastructure compared to larger municipalities [64]. This infrastructure dichotomy is reflected in the risk analysis and is also influenced by the level of information and health education in the respective municipalities.

In more populous municipalities, the majority of animals in public areas are leashed and guided by their owners, indicating a group of animals with better access to prophylactic protocols and waste removal practices [4]. In contrast, in less populated municipalities, the presence of unleased and/or stray animals is more common, which ultimately increases the risk.

## 5. Conclusions

This study highlights the high risk of environmental contamination based on the range and variety of gastrointestinal parasites detected in fecal material, particularly in less populous municipalities. The findings emphasize the potential threat of zoonotic pathogen transmission to the human population in the study regions, focusing on the geohelminths Ancylostomatidae and *Toxocara* spp., as well as the enteric protozoa *Giardia* spp. and *Entamoeba* spp. because of their direct life cycles. *D. caninum*, whose transmission has been linked to the presence of intermediate hosts, was identified as the second most prevalent parasite after hookworms, reinforcing the link between maintenance of the cycle and the risk of human exposure.

The public health threat of infection by zoonotic enteric parasites is increasing, particularly in urban environments. Therefore, all socioeconomic aspects must be considered in the development of public policies for sanitation and health education to devise multidisciplinary approaches based on the One Health concept aimed at reducing the risks associated with environmental contamination by parasites.

## Figures and Tables

**Figure 1 pathogens-14-00211-f001:**
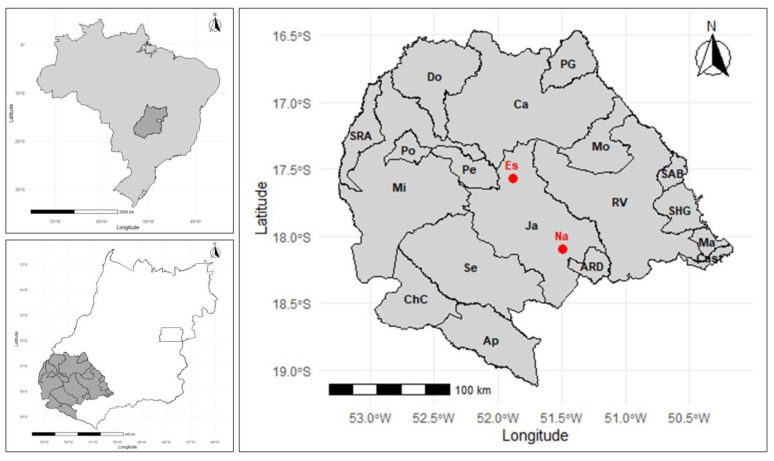
Sample area of municipalities in the southwestern microregion of Goiás, located on the central plateau of Brazil. ARD: Aparecida do Rio Doce; Ap: Aporé; Ca: Caiapônia; Cast: Castelândia; ChC: Chapadão do Céu; Do: Doverlândia; Ja: Jataí; Es: Estância (district of Jataí); Na: Naveslândia (district of Jataí); Ma: Maurilândia; Mi: Mineiros; Mo: Montividiu; PG: Palestina de Goiás; Pe: Perolândia; Po: Portelândia; RV: Rio Verde; SHG: Santa Helena de Goiás; SRA: Santa Rita do Araguaia; SAB: Santo Antônio da Barra; Se: Serranópolis. Source: R Program.

**Figure 2 pathogens-14-00211-f002:**
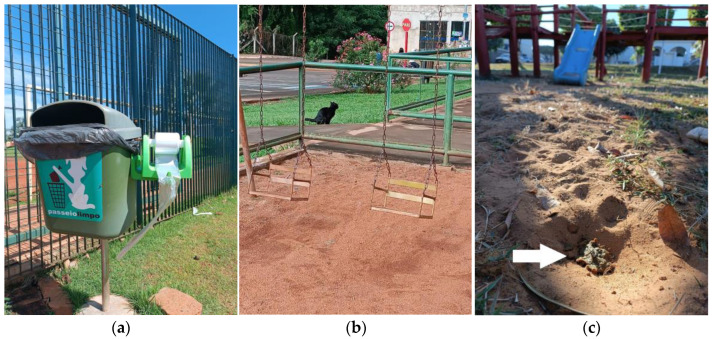
Overview of public squares and parks in the southwest region of Goiás, in the central plateau of Brazil. (**a**) Public awareness policy for animal feces disposal in a public square. (**b**) Domestic cat defecating near a children’s recreational area, which lacks physical containment. (**c**) Fecal material (white arrow) on the ground near the playground.

**Table 1 pathogens-14-00211-t001:** Prevalence of contamination in environmental fecal samples by gastrointestinal parasites, distributed across the municipalities of the southwestern microregion of Goiás, located on the central plateau of Brazil.

Municipality	Inhabitants ^1^	Sampled Areas	Stool Sample	Positive % (CI)
Aparecida do Rio Doce	2907	2	12	12 100 (73.5–100)
Aporé	4325	5	19	13 68.4 (43.5–87.4)
Caiapônia	16,507	8	38	29 76.3 (59.8–88.6)
Chapadão do Céu	12,870	4	16	15 93.8 (69.8–99.8)
Castelândia	2985	5	17	9 52.9 (27.8–77.0)
Doverlândia	6956	3	19	14 73.7 (48.8–90.9)
Jataí	105,729	9	62	50 80.7 (68.6–89.6)
Estância (district)		1	1	1 100 (2.5–100)
Naveslândia (district)		2	7	6 85.7 (42.1–99.6)
Maurilândia	10,304	5	17	9 52.9 (27.8–77.0)
Mineiros	70,081	10	42	31 73.8 (57.9–86.1)
Montividiu	12,521	6	23	19 83.6 (61.2–95.1)
Palestina de Goiás	3132	2	8	6 75.0 (34.9–96.8)
Perolândia	2964	3	18	16 88.9 (65.3–98.6)
Portelândia	3280	2	16	12 75.0 (47.6–92.7)
Rio Verde	225,696	25	133	70 52.6 (43.8–61.4)
Santo Antônio da Barra	4267	3	9	5 55.5 (21.2–86.3)
Santa Helena de Goiás	38,492	12	40	27 67.5 (50.9–81.4)
Santa Rita do Araguaia	5924	5	14	12 85.7 (57.2–98.2)
Serranópolis	8027	5	25	21 84 (63.9–95.5)
	536,973	117	536	37 770.3 (66.3–74.1)

^1^ Number of inhabitants based on IBGE data [17]. CI: confidence interval

**Table 2 pathogens-14-00211-t002:** Prevalence of gastrointestinal parasites identified by municipalities in the southwestern geographical microregion of Goiás, located in the central plateau region of Brazil.

City Code	Ancylostomatidae	*Toxocara* spp.	*Trichuris* spp.	*Strongyloides* spp.	*Dipylidium* *caninum*	*Spirometra* spp.	Taeniidae	*Platynosomum fastosum*	Trematoda	*Giardia* spp.	*Cystosospora* spp.	*Sarcocystis* spp.	*Entamoeba* spp.
ARD	10/12 85.3 (51.6–97.9)	1/12 8.3 (0.2–38.5)	0 -	0 -	9/12 75.0 (42.8–94.5)	0 -	0 -	0 -	0 -	1/12 8.3 (0.2–38.5)	0 -	0 -	1/12 8.3 (0.2–38.5)
Ap	11/19 57.9 (33.5–79.8)	1/19 5.3 (0.1–26.0)	0 -	0 -	8/19 42.1 (20.3–66.5)	0 -	0 -	0 -	0 -	0 -	0 -	0 -	0 -
Ca	25/38 65.8 (48.7–80.4)	4/38 10.5 (2.9–24.8)	0 -	0 -	5/38 13.2 (4.4–28.14)	0 -	0 -	1/38 2.6 (0.1–13.8)	0 -	2/38 5.3 (0.6–17.8)	3/38 7.9 (1.7–21.4)	0 -	0 -
ChC	14/16 87.5 (61.7–98.6)	1/16 6.3 (0.2–30.2)	4/16 25.0 (7.3–52.4)	0 -	0 -	0 -	1/16 6.3 (0.2–30.2)	0 -	0 -	0 -	3/16 18.6 (4.1–45.7)	0 -	1/16 6.3 (0.2–30.2)
Cast	7/17 41.2 (18.4–67.1)	0 -	0 -	0 -	1/17 5.9 (0.2–28.7)	0 -	0 -	0 -	0 -	1/17 5.9 (0.2–28.7)	0 -	0 -	2/17 11.8 (1.5–36.4)
Do	14/19 73.7 (48.8–90.9)	1/19 5.3 (0.1–26.0)	0 -	0 -	4/19 21.1 (6.1–45.6)	0 -	0 -	0 -	0 -	1/19 5.3 (0.1–26.0)	0 -	0 -	0 -
Es	0 -	0 -	0 -	0 -	1/1 100 (2.5–100)	0 -	0 -	0 -	0 -	0 -	0 -	0 -	0 -
Ja	24/62 38.7 (26.6–51.9)	1/62 1.6 (0.1–8.7)	1/62 1.6 (0.1–8.7)	0 -	32/62 51.6 (38.6–64.5)	2/62 3.2 (0.4–11.2)	0 -	1/62 1.6 (0.1–8.7)	0 -	1/62 1.6 (0.1–8.7)	10/62 16.1 (8.0–27.7)	0 -	0 -
Ma	7/17 41.2 (18.4–67.1)	0 -	0 -	0 -	1/17 5.9 (0.2–28.7)	0 -	0 -	0 -	0 -	0 -	0 -	0 -	1/17 5.9 (0.2–28.7)
Mi	28/42 66.7 (50.4–80.4)	0 -	0 -	0 -	11/42 26.2 (13.9–42.1)	0 -	0 -	1/42 2.4 (0.1–12.3)	0 -	1/42 2.4 (0.1–12.3)	3/42 7.1 (1.5–19.5)	0 -	1/42 2.4 (0.1–12.3)
Mo	14/23 60.9 (38.5–80.3)	1/23 4.4 (0.1–21.9)	0 -	0 -	13/23 56.2 (34.5–76.8)	0 -	0 -	0 -	0 -	0 -	1/23 4.4 (0.1–21.9)	0 -	0 -
Na	3/7 42.8 (9.9–81.6)	0 -	0 -	0 -	5/7 71.4 (29.1–96.3)	0 -	0 -	0 -	0 -	0 -	0 -	0 -	1/7 14.3 (0.4–57.9)
PG	4/8 50.0 (15.7–84.3)	2/8 25.0 (3.2–65.1)	0 -	0 -	4/8 50.0 (15.7–84.3)	0 -	0 -	0 -	0 -	0 -	1/8 12.5 (0.3–52.7)	0 -	0 -
Pe	15/18 83.3 (58.6–96.4)	3/18 16.7 (3.6–41.4)	1/18 5.6 (0.1–27.3)	1/18 5.6 (0.1–27.3)	9/18 50.0 (26.0–74.0)	0 -	0 -	0 -	0 -	0 -	2/18 11.1 (1.4–34.7)	1/18 5.6 (0.1–27.3)	0 -
Po	12/16 75.0 (47.6–92.7)	0 -	0 -	0 -	4/16 25.0 (7.3–52.4)	0 -	0 -	0 -	0 -	0 -	2/16 12.5 (1.6–38.4)	0 -	1/16 6.3 (0.2–30.2)
RV	56/133 42.1 (33.6–51.0)	8/133 6.0 (2.6–11.5)	0 -	0 -	11/133 8.3 (4.2–14.3)	0 -	0 -	0 -	1/133 0.8 (0.0–4.1)	9/133 6.8 (3.1–12.5)	3/133 2.3 (0.5–6.5)	0 -	1/133 0.8 (0.0–4.1)
SAB	4/9 44.4 (13.7–78.8)	0 -	1/9 11.1 (0.3–48.3)	0 -	1/9 11.1 (0.3–48.3)	0 -	0 -	0 -	0 -	1/9 11.1 (0.3–48.3)	0 -	0 -	1/9 11.1 (0.3–48.3)
SHG	23/40 57.5 (40.9–73.0)	4/40 10.0 (2.8–23.6)	0 -	0 -	6/40 15.0 (5.7–29.8)	0 -	0 -	0 -	0 -	0 -	1/40 2.5 (0.1–13.2)	0 -	2/40 5.0 (0.6–16.9)
SRA	12/14 85.7 (57.2–98.2)	2/14 14.3 (1.8–42.8)	2/14 14.3 (1.8–42.8)	0 -	4/14 28.6 (8.4–58.1)	0 -	0 -	0 -	0 -	0 -	1/14 7.1 (0.2–33.9)	0 -	1/14 7.1 (0.2–33.9)
Se	20/25 80.0 (59.3–93.2)	4/25 16.0 (4.5–36.1)	0 -	0 -	9/25 36 (18.0–57.5)	0 -	0 -	0 -	0 -	0 -	0 -	0 -	0 -
Total	303/536 56.5 (52.3–60.7)	33/536 6.2 (4.4–8.5)	9/536 1.7 (0.9–3.2)	1/536 0.2 (0–1.1)	138/536 25.8 (22.2–29.6)	2/536 0.4 (0.1–1.4)	1/536 0.2 (0–1.1)	3/536 0.6 (0.2–1.6)	1/536 0.2 (0–1.1)	17/536 3.2 (2.0–5.0)	30/536 5.6 (4.0–7.9)	1/536 0.2 (0–1.1)	13/536 2.4 (1.4–4.1)

ARD: Aparecida do Rio Doce; Ap: Aporé; Ca: Caiapônia; ChC: Chapadão do Céu; Cast: Castelândia; Do: Doverlândia; Es: Estância; Ja: Jataí; Ma: Maurilândia; Mi: Mineiros; Mo: Montividiu; Na: Naveslândia; PG: Palestina de Goiás; Pe: Perolândia; Po: Portelândia; RV: Rio Verde; SAB: Santo Antônio da Barra; SHG: Santa Helena de Goiás; SRA: Santa Rita do Araguaia; Se: Serranópolis.

## Data Availability

Data are contained within the article.

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
