# Peer review of "Risk of Environmental Contamination by Gastrointestinal Parasites in Public Areas of the Central Plateau Microregion of Brazil: A Public Health Concern"

_pathogens, 2025, doi:10.3390/pathogens14030211_

Round 1
Reviewer 1 Report
Comments and Suggestions for Authors
Dear Authors,
First of all, I would like to congratulate everyone involved in this study. It is not common to find manuscripts of this quality in a first review. You have made my work as a reviewer simple and efficient. Below, I outline a series of minor errors and considerations for you to correct, which I believe will not take much of your time. Once again, congratulations on your work.
L47: I believe that an expression like "living as strays" would be better than "abandoned."
L65: The language should be corrected. This title is written in Portuguese.
L81: Coproparasitological (not co-proparasitological).
L86-91: I cannot locate the municipalities of Estância, Naveslândia, and Santa Helena on the map. Likewise, the acronym SHG does not correspond to any municipality listed.
L105: form of a parasite
L179-180: Please provide a reference to support this statement.
L193: The authors do not specify the sample collection dates. I am unaware if environmental conditions between the different sampling areas and collection dates could affect the viability of parasitic forms, leading to under-detection. Please elaborate on this point.
L206: Since this study does not detect Ancylostoma ceylanicum and it is an allochthonous species, I do not believe it is appropriate to mention it specifically (although I do find it correct to mention A. braziliense and A. caninum, which are native species).
L253: Since these percentages come from another study, I believe it is more appropriate to mention only that genotypes A and B have been linked to human infections, without specifying exact figures.
L274: Please provide a reference to justify this statement.
Comments on the Quality of English Language
The comments regarding English language errors have already been communicated to the authors in the general review.
Author Response
Reviewer 1 (changes requested in yellow)
Comments: Dear Authors, First of all, I would like to congratulate everyone involved in this study. It is not common to find manuscripts of this quality in a first review. You have made my work as a reviewer simple and efficient. Below, I outline a series of minor errors and considerations for you to correct, which I believe will not take much of your time. Once again, congratulations on your work.
Response: We are very flattered and grateful for the comment. Indeed, as researchers we are always trying to improve, and we are happy to see that our efforts were successful. We also appreciate every consideration given, as it helps us in the process of improving and bringing the best possible version of our study.
Comments: L47: I believe that an expression like "living as strays" would be better than "abandoned.".
Response: Changed as requested (L48).
Comments: L65: The language should be corrected. This title is written in Portuguese.
Response: Changed as requested (L76).
Comments: L81: Coproparasitological (not co-proparasitological).
Response: Changed as requested (L98).
Comments: L86-91: I cannot locate the municipalities of Estância, Naveslândia, and Santa Helena on the map. Likewise, the acronym SHG does not correspond to any municipality listed.
Response: We corrected the figure by inserting Naveslandia and Estância, which are not municipalities, but districts (this observation was pointed out in the caption). We also inserted the reference to the acronym SHG, which in fact stands for Santa Helena de Goiás (L104-108).
Comments: L105: form of a parasite.
Response: Changed as requested (L121).
Comments: L179-180: Please provide a reference to support this statement.
Response: Thank you for your consideration. We really didn't notice the absence that was inserted as requested (L189).
Comments: L193: The authors do not specify the sample collection dates. I am unaware if environmental conditions between the different sampling areas and collection dates could affect the viability of parasitic forms, leading to under-detection. Please elaborate on this point.
Response: The collections were carried out in several campaigns throughout the year and for more than one year, due to the territorial extension. We state what is stated in the text calmly, because the study area is not an area with noticeable seasonal variations. We literally do not have well-defined Summer, Autumn, Winter and Spring. Only two hot seasons with variations in humidity. However, we explain better, inserting the following excerpt in the text (L221-225):
Although our collections occurred at different times of the year, due to the territorial extension covered, Brazilian Cerrado is a biome characterized by little impact by seasonal changes as in temperate zone biomes. It is noted that the temperature variation is small [16,17] compared to areas where we clearly perceive summer, autumn, winter and spring.
Comments: L206: Since this study does not detect Ancylostoma ceylanicum and it is an allochthonous species, I do not believe it is appropriate to mention it specifically (although I do find it correct to mention A. braziliense and A. caninum, which are native species).
Response: We believe that it is probably A. caninum or A. brasiliense. However, we made the citation because the study (reference number 40) mentions the occurrence of A. ceylanicum from Colombia in South America and also addresses a caution regarding its adaptation in other countries, even though the authors point out that despite not being endemic, there are records including in Brazil. Therefore, we chose to only leave the mention, as a piece of information for caution and future investigations.
Comments: L253: Since these percentages come from another study, I believe it is more appropriate to mention only that genotypes A and B have been linked to human infections, without specifying exact figures.
Response: Changed as requested (L289).
Comments: L274: Please provide a reference to justify this statement.
Response: Changed as requested (L316-317)
Reviewer 2 Report
Comments and Suggestions for Authors
It is a good job
There is not many references to this kind of researches in Brazil or in other countries. you should mention it in the text.
I am missing during the text the origin of the samples. Are from domestic animals? Is there any chance of wild animals feces? is there any chance of human origin? Maybe you should mention at least if they were from cats, dogs or whatever...or at least to make differences between domestic or wild animals.
introduction: Is there any chance of interaction of wildlife in the diferent parks selected? are they fenced? It would be interesting if you have both kind of parks. Maybe you can find interesting results.
I would give some information about the prevalence of these pathogens in Brazil or in your researching area or different parts of the country.
2.1 (translate into English)
I can suggest you to describe the sampling methods. When did you take the samples? How often did you make the sampling? Every morning durint the sampling period? Do you have any idea of the species sampled? If you have any idea of the sampling (cats always have the same place) you could make an specific selection of pathogens.
2.3: you should mention that you made an risk factors analysis, with Chi-square test...and later, in results you could give the result of population size as a risk factor. Did you take any other factors?
3.3. you should mention other risk factors that had no statistically significant correlation. Why did you choose this and no other elements to make this test? This should be explained.
4.1. line154: why are you comparing the extension here? you could give this details in the intro or are there any other research about the same pathogens?
line 161: This also should be in the intro. You can justify the research and also the results. Maybe in this paragraph you could include the origin of the feces (domestic, abandoned, wild animals).
You talk about Trichuris vulpis but again...you do not mention the possible interaction between domestic or wildlife or human. Is there any interface between wild and domestic animals?
Figure 2: do the parks have any pannels about leaving feces? do the government punish if the owners leave feces?
I miss some researches of the selected pathogens in other areas/countries to compare the results with yours. This could give more importance to your research.
Giardia: do you have any data of the global prevalence of giardia in humans? prevalence in Brazil?
You should give more importance to the risk factors. maybe you could say some hypothesis and extend the discussion here (line 279).
Do not you have any other research to continue with this subject? It is really interesting to follow the line or also to get deeper researches. you should mention it in the conclusions: the need of continue and investigate other parks, areas or species.
Author Response
Reviewer 2 (changes requested in green)
Comments: It is a good job. There is not many references to this kind of researches in Brazil or in other countries. you should mention it in the text.
Response: Dear reviewer, thank you for your considerations and for the opportunity to improve our study. We are very grateful. We have inserted the mention reference in the text (L56).
Comments: I am missing during the text the origin of the samples. Are from domestic animals? Is there any chance of wild animals feces? is there any chance of human origin? Maybe you should mention at least if they were from cats, dogs or whatever...or at least to make differences between domestic or wild animals.
Response: We cannot necessarily guarantee that the feces are from domestic animals or wild animals. However, any statement about this would be speculative. Wild animals are not found in these areas, except in very rare cases when a stray animal invades the city. Therefore, we find it difficult to make a definitive statement without being speculative and so we treat the results only as environmental contamination. However, we insert the following observation in the methods (L95-96):
The samples were selected when they were similar to dog and cat feces. We also observed the places where dogs and cats usually defecate in the squares.
Comments: Introduction: Is there any chance of interaction of wildlife in the diferent parks selected? are they fenced? It would be interesting if you have both kind of parks. Maybe you can find interesting results.
Response: As we mentioned in the previous answer, these parks do not have records of the presence of wild animals, except in very rare exceptions and cases of lost wild animals. However, as a rule, we consider the samples as dogs and cats, because the places where they were found were places where domestic animals were frequently seen. The parks are in highly anthropized areas and are parks intended for human physical activities and recreation. We inserted the term recreational in L86 to improve the identification of the purpose of the parks studied.
Comments: I would give some information about the prevalence of these pathogens in Brazil or in your researching area or different parts of the country.
Response: We have inserted the following excerpt (L49-51):
Studies in Brazil and around the world demonstrate a prevalence of gastrointestinal parasites in dogs of 16.1-62.2% and in cats of 3.3-90.9% according to Souza et al. [3].
Comments: 2.1 (translate into English).
Response: Changed as requested (L76). This change is in yellow because it was also verified by reviewer 1.
Comments: I can suggest you to describe the sampling methods. When did you take the samples? How often did you make the sampling? Every morning durint the sampling period? Do you have any idea of the species sampled? If you have any idea of the sampling (cats always have the same place) you could make an specific selection of pathogens.
Response: As we mentioned earlier, the samples were probably from dogs and cats and this was included in the text, but needing to segment the results by animal would be very speculative and we thought of an analysis more focused on environmental contamination, since environmental health is neglected in several studies. Regarding the collection periods, we included greater detail according to L88-91.
Comments: 2.3: you should mention that you made an risk factors analysis, with Chi-square test...and later, in results you could give the result of population size as a risk factor. Did you take any other factors?
Response: We removed the classification of cities from item 2.3 as suggested. Unfortunately, we did not perform another risk factor analysis, as we considered that only population size was appropriate, which was a precise piece of data that we could obtain for each municipality. We even considered comparing humidity and temperature, but in addition to this being more appropriate for seasonal collections throughout the year, few of these municipalities have this information accurately.
Comments: 3.3. you should mention other risk factors that had no statistically significant correlation. Why did you choose this and no other elements to make this test? This should be explained.
Response: We explained this in the previous answer. We did not find a possible and necessary way to write this in the text. We are very grateful for your consideration and hope to have answered this question. If not, we are available for new suggestions.
Comments: 4.1. line154: why are you comparing the extension here? you could give this details in the intro or are there any other research about the same pathogens?
Response: In fact, we considered this comparison to introduce the discussion and show the relevance of the study in comparison to other countries. We did not know what the reviewers' position and opinion regarding the study would be, and we did not want the regional aspect to be highlighted, so we thought that introducing the discussion by showing the size of the studied area is fundamental for understanding not only the reviewers but also the readers of the study's relevance.
Comments: line 161: This also should be in the intro. You can justify the research and also the results. Maybe in this paragraph you could include the origin of the feces (domestic, abandoned, wild animals).
Response: Thank you very much, We agree with you on this paragraph, we have moved the paragraph to the introduction and inserted the information about the origin of the samples, as we detailed in previous answers, in the materials and methods (L60-66).
Comments: You talk about Trichuris vulpis but again...you do not mention the possible interaction between domestic or wildlife or human. Is there any interface between wild and domestic animals?
Response: We believe that this issue has already been addressed in previous questions and the absence of this interface with wild animals in the urban environment has been clarified.
Comments: Figure 2: do the parks have any pannels about leaving feces? do the government punish if the owners leave feces?
Response: Unfortunately, there is no government order for feces to be collected and thrown in the trash and consequently no punishment or penalty. We add this in lines 197-199.
Comments: I miss some researches of the selected pathogens in other areas/countries to compare the results with yours. This could give more importance to your research.
Response: Perfect suggestion. We insert it into the text as per lines 200-215.
Comments: I miss some researches of the selected pathogens in other areas/countries to compare the results with yours. This could give more importance to your research.
Response: Perfect suggestion. We insert it into the text as per lines 200-215.
Comments: Giardia: do you have any data of the global prevalence of giardia in humans? prevalence in Brazil?
Response: We insert it into the text as per lines 283-285.
Comments: You should give more importance to the risk factors. maybe you could say some hypothesis and extend the discussion here (line 279).
Response: Perfect suggestion. We insert it into the text as per lines 289-296.
Comments: Do not you have any other research to continue with this subject? It is really interesting to follow the line or also to get deeper researches. you should mention it in the conclusions: the need of continue and investigate other parks, areas or species.
Response: We are immensely grateful for your time and for analyzing our study with such care. We had the opportunity to greatly improve our studies. We have worked extensively with data on zoonotic pathogens in domestic animals and also in wild animals from roadkill. Our line of research has been solidified in our region. In total, six manuscripts have been published in recent years, in which we have identified, in addition to zoonotic risks, approaches to improve the ways of understanding parasitic diseases for diagnostic and epidemiological purposes. Regarding the continuity of this work, we intend to develop an environmental education project with children, mainly from schools with children in socioeconomic vulnerability. We preferred not to address these aspects in the conclusion, as we still face some challenges in funding research in Brazil, but we assure you that we will continue to try and insist on research to change some realities in Brazil.
Round 2
Reviewer 1 Report
Comments and Suggestions for Authors
Dear Authors,
I am reaching out to you once again to congratulate you on your research and to point out a few minor errors in the new version of your manuscript:
The following scientific names should be written in italics:
Toxocara (L202, 205).
Echinococcus (L207).
Taenia (L207).
Giardia (L287 and 289).
That is all from my side. Best regards.
Author Response
Dear Reviewer, thank you very much for each suggested change. All the words have been arranged as suggested.
Best Regards.